# The Influence of Titanium Oxide Nanoparticles and UV Radiation on the Electrical Properties of PEDOT:PSS-Coated Cotton Fabrics

**DOI:** 10.3390/ma16041738

**Published:** 2023-02-20

**Authors:** Fahad Alhashmi Alamer, Rawan F. Beyari

**Affiliations:** Department of Physics, Faculty of Applied Science, Umm AL-Qura University, Al Taif Road, Makkah 24382, Saudi Arabia

**Keywords:** conductive cotton, PEDOT:PSS, TiO_2_, nanocomposite, steam washing, metallic

## Abstract

With the rapid growth of electronic textiles, there is a need for highly conductive fabrics containing fewer conductive materials, allowing them to maintain flexibility, low cost and light weight. Poly(3,4-ethylenedioxythiophene): polystyrene sulfonate (PEDOT:PSS), is one of the most promising conductive materials for the production of conductive fabrics due to its excellent properties such as solubility, relatively high conductivity, and market availability. Moreover, its electrical conductivity can be enhanced by polar solvents or acid treatment. The aim of this work was to fabricate conductive cotton fabrics with a small fixed amount of PEDOT:PSS and to investigate how titanium dioxide (TiO_2_) nanoparticles affect the electrical, thermal and structural properties of PEDOT:PSS-coated cotton fabrics. The change in electrical conductivity of the nanocomposite fabric was then related to morphological analysis by scanning electron microscopy and X-ray diffraction. We found that the sheet resistance of the nanocomposite cotton fabric depends on the TiO_2_ concentration, with a minimum value of 2.68 Ω/□ at 2.92 wt% TiO_2_. The effect of UV light on the sheet resistance of the nanocomposite cotton fabric was also investigated; we found that UV irradiation leads to an increase in conductivity at an irradiation time of 10 min, after which the conductivity decreases with increasing irradiation time. In addition, the electrical behavior of the nanocomposite cotton fabric as a function of temperature was investigated. The nanocomposite fabrics exhibited metallic behavior at high-TiO_2_ concentrations of 40.20 wt% and metallic semiconducting behavior at low and medium concentrations of 11.33 and 28.50 wt%, respectively. Interestingly, cotton fabrics coated with nanocomposite possessed excellent washing durability even after seven steam washes.

## 1. Introduction

Recently, nanotechnology has played a significant role in the development of smart textiles by improving their physical and chemical properties and increasing their durability [1]. In addition, the use of nanomaterials in the production of smart textiles imparts other important properties to fabrics, such as antibacterial properties [2], protection against UV radiation [3], and water-repellent properties [4], without altering the breathability, flexibility, lightness, and tactility of the fabrics. Smart textiles treated with nanoparticles have a wide range of applications in electronics [5], environment [6], sports [7], human reproduction [8], pharmaceuticals [9], and medicine [10].

Cotton fabrics are ideal for the creation of smart textiles because they are very comfortable to wear, have high absorbency and hydrophobicity, are inexpensive, and are resistant to static electricity [11]. In general, conductive cotton fabrics can be easily produced using three methods [12]: infusion with nanomaterials [13], coating with conductive polymers [14], or coating with conductive polymers and nanomaterials [15].

The first method involves incorporating nanomaterials into the fabric substrates, which is believed to affect the physical properties of the substrates. Nanomaterials such as silver nanoparticles, zinc oxide nanoparticles, and copper oxide nanoparticles are commonly used as conductive fillers for functionalized conductive fabrics. Titanium nanoparticles have attracted significant attention as conductive fillers for the production of conductive fabrics due to their interesting optical, electrical, and photocatalytic properties, as well as their low cost, safety, ability to filter impurities, chemical stability, and non-toxicity [16]. Perelshtein et al. [17] applied titanium dioxide (TiO_2_) nanoparticles to cotton fabrics and then exposed them to UV radiation. The results show that the modified cotton had antimicrobial properties because the nanoparticles became more stable. Two types of modified cotton fabrics with TiO_2_ nanoparticles were also prepared using three different fabrication methods; namely, sonochemical, hydrothermal, and solvothermal methods [18]. The modified cotton fabrics exhibited self-cleaning properties, as methylene blue was 99% degraded within 60 min after being exposed to sunlight. In addition, the modified samples exhibited antibacterial properties against *E. coli* and *B. pumilus*, and these properties were found to depend on the type of cotton. In [19], nanocomposites containing TiO_2_ nanoparticles and chitosan were used to produce cotton fabrics. Compared to pure cotton, an improvement in UV protection and antibacterial properties was observed, with the percentage of bacterial reduction increasing up to 99.8% against *E. coli* and 97.3% against *S. aureus* [20]. In another study [21], TiO_2_ nanoparticles and graphene oxide were deposited on cotton fabric via an immersion and drying process. The cotton treated with TiO_2_ nanoparticles and graphene oxide exhibited self-cleaning behavior after the samples were exposed to sunlight and became electrically conductive compared to untreated cotton and cotton treated only with graphene oxide. Recently, Naggar et al. [22] prepared cotton fabric with TiO_2_ nanoparticles, silver nanoparticles, and a mixture of both using the pad-dry-cure method. The cotton modified with the mixture of nanoparticles showed high antimicrobial activity and UV protection with excellent mechanical properties compared to the cottons modified with TiO_2_ nanoparticles and silver nanoparticles.

A second known method for the production of functionalized conductive fabric substrates involves using conductive polymers, which are commercially available, durable, flexible, and easy to make. This method not only imparts conductivity to the fabrics, but also allows them to retain their mechanical properties. PEDOT:PSS is an important conductive polymer from which extremely conductive and flexible smart fabrics can be made [23,24,25]. In the literature, different methods have been used to coat fabrics with PEDOT:PSS, such as drop casting, dip coating, spraying, etc. [26,27]. A highly conductive cotton fabric with a sheet resistance of 1.58 Ω/□ was obtained when a concentration of 21.7 wt% PEDOT:PSS doped with dimethyl sulfoxide (DMSO) was used to prepare a conductive cotton fabric. In another study [28], the type of cotton fabrics was found to have an effect on the electrical conductivity of cotton fabrics after treatment with PEDOT:PSS with excellent electrical conductivity. In another study [29], cotton fabrics treated with DMSO-doped PEDOT:PSS ink were used as ECG electrodes. The ECG electrodes showed high efficiency, achieving excellent signals without skin irritation. In a study by Alamer et al. [16], the type of cotton fabric treated with PEDOT:PSS affected the electrical conductivity, and the treated cotton fabrics showed metallic behavior when the samples were heated from room temperature to 100 °C. In a recent study [30], a Joule heater was fabricated using PEDOT:PSS-based cotton fabrics, applying different DC voltage values to the fabrics and measuring the surface temperature within one minute. The maximum heating temperature of the treated fabric was 94 °C at an applied voltage of 40 V.

A third method for fabricating conductive fabric substrates involves the combination of PEDOT:PSS as a conductive material and nanomaterials as an aid to improve electrical conductivity [31]. These fabrics can be used in various applications such as energy storage, power generation, UV blocking, and for antibacterial purposes [32]. Alamer et al. [33,34] fabricated conductive cotton fabrics using a nanocomposite of PE-DOT:PSS and single-walled carbon nanotubes. The results showed that the electrical conductivity of the conductive cotton fabrics depended on the fabrication method, and the conductive cotton fabrics fabricated from a layer of SWCNTs-PEDOT:PSS-SWCNTs exhibited high electrical conductivity. The conductive cotton threads were also fabricated using PEDOT:PSS alone, single-walled carbon nanotubes alone, and a nanocomposite of both [35] The results showed that the conductive thread prepared with the nanocomposite exhibited high electrical conductivity compared to the PEDOT:PSS thread and the thread made of single-walled carbon nanotubes. In a recent study [36], a nanocomposite of PEDOT:PSS and multi-walled carbon nanotubes was used to fabricate conductive cotton fabrics. The results showed that the sheet resistance of the conductive cotton fabrics depended on the concentration of the multi-walled carbon nanotubes at a fixed amount of PEDOT:PSS and reached the minimum value of 2.86 Ω/□ at a concentration of 60.27 wt%.

Otley et al. [37] used PEDOT:PSS to coat the surface of PET fabrics whose surfaces contained silica nanoparticles. The concentration of PEDOT:PSS affected the electrical conductivity of the conductive fabric, with a minimum sheet resistance of 3.2 Ω/□ and a high current-carrying capacity. The conductive fabrics exhibited metallic behavior. This high conductivity was due to the presence of silica nanoparticles, which led to phase separation between PEDOT and PSS on the surface of the fabrics, increasing the electrical conductivity. In another study [38], silver nanoparticles were immobilized on a cotton fabric and the fabric was then coated with PEDOT:PSS using mist polymerization to prevent the substances from detaching and forming cracks. The sheet resistance and electromagnetic shielding of the conductive fabric were 8.7 Ω/□ and 27.1 dB, respectively. In a study by Anbalagan et al. [39], the effect of gamma radiation on the electrical properties of PEDOT:PSS was investigated. The results showed that the structure of PEDOT:PSS did not change after gamma ray irradiation, but the electrical conductivity and mobility decreased with increasing gamma ray dose. In another study [40], PEDOT:PSS films were irradiated with UV light at different irradiation times. The results showed that UV light affected the structure of PEDOT:PSS in a specific PEDOT:PSS chain, which affected the electrical conductivity of PEDOT:PSS films.

The purpose of this study was to produce highly electrically conductive cotton fabrics with a lower amount of conductive materials by concentration control of TiO_2_ nanoparticles and a fixed amount of PEDOT:PSS. The concentration control of the nanocomposite was realized by the drop casting method followed by drying. This method was chosen because it is a straightforward procedure that does not lose materials and does not require the use of special tools. Using this method, we fabricated conductive cotton fabrics with a wide range of sheet resistances, with a minimum value of 2.68 Ω/□ for a nanocomposite containing TiO_2_ at 2.92 wt% in the polymer matrix. The effects of UV light irradiation and temperature on the electrical conductivity of the cotton fabric coated with the nanocomposite were investigated. In addition, the activation energies were calculated based on the Arrhenius equation. The morphological, structural, and thermal properties of the samples were characterized using SEM, FTIR, XRD, TGA, and DTA analyses. The electrical properties of the samples were also investigated. As a proof of concept, the nanocomposite cotton fabric was steam-washed for up to seven cycles and sheet resistance and mass loss were evaluated to test the durability of the cotton fabric.

## 2. Experimental Section

### 2.1. Materials

The cotton fabrics were obtained from FPC, SA, and used as received. The aqueous solution of PEDOT:PSS with high conductivity was obtained from Sigma Aldrich, Gillingham, UK (pH ≤ 2; solids content ≈ 1–1.2 wt%; viscosity 30–100 cps). The resistivity of PEDOT:PSS according to the product specification is ≤100 Ω/□. The water solubility and the density of the product are 0.06 g/L and 0.999 g/mL at 25 °C, respectively. Dimethyl sulfoxide (DMSO) was obtained from Sigma Aldrich, UK, and used as received. Titanium dioxide (TiO_2_) nanoparticles were also purchased from Sigma Aldrich, UK, (particle size less than 100 nm). The physical state of TiO_2_ nanoparticles is powdery, has a white color and is odorless. Its water solubility and density are 0.001 g/L at 20 °C and 4.26 g/mL at 25 °C, respectively. Mechiakha et al. [41] investigated the microstructure and the optical properties of TiO_2_. The TiO_2_ films were first deposited on silicon and sapphire substrates using the sol-gel method. The results showed that the TiO_2_ films were uniform, and their crystallinity, density, and roughness increased with increasing annealing temperature, as revealed by AFM analysis. The transparency of the TiO_2_ films was high and decreased with increasing annealing temperature, as shown by the UV-vis spectrometer.

### 2.2. Fabrication of Conductive Cotton

First, two solutions were prepared as follows: The first solution (reference solution) was prepared by mixing 5 wt% DMSO with the aqueous PEDOT:PSS solution and then sonicated at 24 °C for 10 min. This solution was used to prepare the reference sample (cotton fabric coated with PEDOT:PSS) and to prepare the nanocomposite solution. The second solution (nanocomposite solution) was prepared by adding the desired TiO_2_ concentration to the reference solution. Then, the solution was magnetically attracted at 24 °C for 30 min, which confirmed the PEDOT:PSS/TiO_2_ nanocomposite solution.

Subsequently, the cotton fabrics were cut to the same area (2.5 × 2.5 cm^2^) and the drop-casting and drying process was used to prepare the cotton fabric coated with PEDOT:PSS (reference sample) and the nanocomposite of cotton fabrics treated with PEDOT:PSS/TiO_2_. The solutions were dropped onto the cotton fabrics, stored for 30 min and dried in an oven at 100 °C for one hour. A series of conductive cottons was prepared using this method, with each sample having a different TiO_2_ concentration (see Figure 1). The concentration of PEDOT:PSS in the reference sample and nanocomposite samples was determined to be 9.11 wt%, and the concentrations of TiO_2_ in the nanocomposite-treated cotton fabrics ranged from 2.22 to 13.29 wt%.

### 2.3. Characterization Method

*Sheet resistance measurements*: The electrical properties of the cotton fabrics coated with PEDOT:PSS (reference sample) and PEDOT:PSS with different TiO_2_ concentrations were investigated using the four-probe technique as described in the literature [42] and homemade probes consisting of four evenly spaced copper probes (0.35 cm). Current from the outer lead was applied and measured using a Peak Toch 3340 DMM multimeter, and the differential voltage from the inner lead was recorded using the same tool. The electrical resistance (*R*) was determined from an *I-V* curve; then, the sheet resistance (*R_s_*) was calculated in the unit of Ω/□ using the relationship *R_s_ = R(w*/*l*), where *w* is the sample width (2.5 cm) and *l* is the distance between the leads (0.35 cm).

*Resistance as function of temperature*: To determine how the treated sample behaves under the influence of temperature, i.e., whether it behaves as a semiconductor, as a metal, or as both, the electrical resistance was obtained in the range 30 to 100 °C, as explained in the previous section. This experiment was performed using a furnace with digital temperature control.

*UV-light irradiation:* The effect of UV light irradiation on the sheet resistance of the treated cotton fabric was studied using a small chamber, where the cotton fabrics were irradiated with UV light (λ = 365 nm) at a distance of 20 cm. The cotton fabrics were irradiated for periods of 10 min to 180 min.

*Morphological analysis*: The surface morphology and elemental composition of the samples were determined using a high-power scanning electron microscope (Hitachi Japan, model-S3400N) with a resolution of 3.0 nm in high vacuum mode. Images of the untreated and treated samples were obtained without coating the samples with conductive materials, compared to most conventional methods. The instrument was equipped with an energy-dispersive X-ray spectrometer (EDS).

*XRD analysis*: XRD patterns of all samples were obtained using a Rigaku (Japan, Smart Labs model) with core dimensions of 1300 × 1880 × 1300 mm^3^. The generator was operated at 40 kV and 100 mA, and intensities were measured in the range of 2θ from 4° to 70°. The measurements were performed with a scan rate of 0.02/s and a slit width of 0.1 mm using a Cu–Ka radiation source (λ = 1.540598 Å).

*FTIR analysis*: The FTIR spectra of the untreated cotton, PEDOT:PSS-treated cotton, and nanocomposite were recorded using a PerkinElmer FTIR analyzer (USA) under ambient conditions in the wavenumber range 4000–500 cm^−1^.

*Thermal analysis*: Thermogravimetric analysis (TGA) and differential thermal analysis (DTA) were performed on the untreated and PEDOT:PSS-treated cotton and on the nanocomposite using the simultaneous instrument TG/DTA model DTG-60 under a dynamic nitrogen atmosphere, with a heating rate of 50 °C/min from 30 °C to 900 °C.

*Washability:* The effect of steam washing on the electrical performance of the treated cotton fabric was investigated using a steam washing machine (Babyliss, China; V = 220 V–240 V; frequency ~50–60 Hz; power = 1260–1500 W). The steam washing procedure was repeated for seven cycles, with each cycle lasting 20 min.

## 3. Results and Discussion

### 3.1. Effect of TiO_2_ on the Electrical Properties

The sheet resistance of the cotton fabric prepared with 9.11 wt% PEDOT:PSS was initially measured and found to be 20.71 Ω/□. This sample is referred to as the reference sample, i.e., the sample without TiO_2_. Five samples with different concentrations of TiO_2_ (2.22, 2.92, 3.39, 5.29, and 13.29 wt%) were prepared with a fixed amount of PEDOT:PSS of 9.11 wt%; then, the sheet resistance of each sample was measured at room temperature. For the nanocomposite sample prepared with 2.22 wt% TiO_2_, the sheet resistance of the conductive cotton decreased by 81.41% from 20.71 Ω/□ to 3.85 Ω/□, as shown in Table 1. A slight increase in TiO_2_ concentration to 2.92 wt% resulted in a decrease in sheet resistance to the minimum value of 2.68 Ω/□. We considered this concentration as the optimum concentration in the polymer matrix, as it showed the lowest sheet resistance, indicating high electrical conductivity of the PEDOT:PSS cotton fabric with TiO_2_ due to the metallic properties of TiO_2_. However, further increasing the TiO_2_ concentration seems to increase the sheet resistance value, which might be due to particle–particle aggregation of the nanoparticles, as we will discuss in the SEM analysis. Table 2 compares our results and other reports of the literature.

### 3.2. Effect of UV Irradiation on the Electrical Properties

The effect of UV light irradiation on the electrical properties of cotton fabrics impregnated with the nanocomposite PEDOT:PSS/TiO_2_ was studied, focusing particularly on sheet resistance. The sample treated with the formulation 0.5 mL PEDOT:PSS+ 5 wt% DMSO+ 10 mL TiO_2_ was prepared using the same procedure described in Section 2.2. Then, the sheet resistance of the nanocomposite sample was measured as 11.87 Ω/□; this sheet resistance is referred to as the sheet resistance before UV irradiation (t = 0). The dependence of the sheet resistance of the sample on irradiation time is shown in Figure 2. When the nanocomposite sample was irradiated with UV light for 10 min, the sheet resistance of the nanocomposite sample decreased by 53.41% from 11.87 Ω/□ to 5.53 Ω/□. We considered this irradiation time as the optimum time, as it led to the lowest sheet resistance, indicating high electrical conductivity of the nanocomposite cotton fabric. This high electrical conductivity is due to the decomposition of the organic chemical bonds in the PEDT:PSS structure, whose binding energy is smaller than the energy of 365 nm UV light (327.74 kJ/mol). The energy (*E*) was calculated using the relationship $E=\frac{Nhc\times 105}{\lambda }$, where *N* is Avogadro’s number, *h* is Planck’s constant, *c* is the speed of light, and *λ* is the wavelength. Table 3 shows the binding energy of some chemical bonds in the PEDOT:PSS structure. However, with an increasing UV irradiation time of more than 10 min, the sheet resistance of the nanocomposite sample increased linearly by 253.42%, reaching a maximum value of 41.91 Ω/□ at an irradiation time of 180 min, indicating a decrease in electrical conductivity due to the excitation of electrons from the valance band to the conduction band in TiO_2_. These electrons caused a de-doping process as they were transferred from TiO_2_ (n-type semiconductor) to PEDOT:PSS (p-type conductive polymer). As a result, PEDOT:PSS can easily be de-doped, which increases sheet resistance and decreases conductivity [46].

### 3.3. Electrical Conductivity as a Function of Temperature

The electrical conductivity of nanocomposite cotton fabric at low, medium, and high concentrations of TiO_2_ in the temperature range 30 to 100 °C was investigated. The electrical conductivity was calculated using the equation $\;\sigma ={\left(\frac{\pi wRF_{1}F_{2}}{ln\;2}\right)}^{-1}$, where *w* is the width of the sample (2.5 cm), *R* is the measured resistance, and the geometric factor $F_{1}.F_{2}=1$ because $\frac{w}{d}\gg 1$, where *d* is the distance between the probes. As shown in Figure 3A, the nanocomposite samples with low and medium concentrations exhibited two behaviors that strongly depended on the respective temperature. First, a semiconducting behavior was observed in which conductivity increased with increasing temperature; then, at 60 °C, a transition to metallic behavior occurred, where the conductivity decreased with increasing temperature. The nanocomposite sample with high TiO_2_ concentration showed metallic behavior over the entire temperature range. The decrease in conductivity during heating indicates that the electrical conduction paths are disturbed as the temperature increases, which is probably due to the thermal expansion of the composites. Comparison between the three samples also shows that the conductivity during the heating process depends on the amount of TiO_2_ in the nanocomposite, with the conductivity of the nanocomposite with high TiO_2_ decreasing more than that of the nanocomposites with medium and low TiO_2_ [47].

Conductivity as a function of temperature can be analyzed using the Arrhenius equation $\sigma =\sigma_{0}e^{\frac{-E_{a}}{kT}}$, where $\sigma_{0}$ is the pre-exponential factor, $E_{a}$ is the activation energy for hopping (eV), T is the temperature (K), and *k* is Boltzmann’s constant ($8.617\times 10^{-5}\mathrm{eV}$). The plots between lnσ and $\frac{1000}{T}$ for the nanocomposite samples are shown in Figure 3B to 3D. For the low and intermediate concentrations, it was observed that lnσ increases linearly with $\frac{1}{T}\;;\;$then, a transition occurs at 60 °C, where lnσ decreases linearly with $\;\frac{1}{T}$, indicating positive and negative activation energies, as shown in Table 4. However, in the high nanocomposite sample, lnσ decreases linearly with $\frac{1}{T}$ over the entire temperature range, with the activation energy calculated to be about −0.3135 eV.

### 3.4. Morphological Characterization

Figure 4 shows the morphological analysis of the untreated and nanocomposite-treated cotton fabrics under the scanning electron microscope. The SEM image of the untreated cotton fabric (Figure 4A) shows smooth fibers in the longitudinal direction with some fibers twisted back; there are also spaces between the fibers. The average thickness of the fibers is 7.27 μm. The SEM image of the cotton fabric treated with PEDOT:PSS (Figure 4B) shows that the PEDOT:PSS films cover the fibers and fill some of the interstices, which explains why the sample became conductive with a sheet resistance of 20.71 Ω/□, as the polymer films created a conduction path. Figure 4C,D show SEM images of the PEDOT:PSS/TiO_2_ nanocomposite at low and high TiO_2_ concentrations, respectively, demonstrating the incorporation of TiO_2_ particles into the polymer matrix. The high magnification in Figure 3C shows the uniform distribution of TiO_2_ particles on the PEDOT:PSS films without agglomeration of nanoparticles. This concentration—2.29 wt%—represents the optimum concentration introduced into the polymer matrix and gives the lowest sheet resistance, indicating high electrical conductivity. When the TiO_2_ concentration increased to a high level, the TiO_2_ particles agglomerated, which can be clearly seen in Figure 4D and could explain the increase in sheet resistance from 2.68 to 13.57 Ω/□, which indicates a decrease in conductivity [48].

### 3.5. Elemental Analysis

Energy dispersive X-ray spectral analysis (EDS) of untreated, PEDOT:PSS-treated, and nanocomposite-treated cotton was performed to determine the elements on the surface, as shown in Table 5 and Table 6. Carbon and oxygen were observed in the EDS analysis of the untreated cotton fabric at 0.277 and 0.525 keV due to the cellulose structure. In the EDS analysis of the cotton fabric coated with PEDOT:PSS, sulfur was detected at 2.307 keV, mainly from PEDOT:PSS, in addition to carbon and oxygen. In addition, the mass and atomic fractions of carbon were increased in the PEDOT:PSS-coated cotton fabric compared to the untreated cotton fabric. In the EDS analysis of the cotton fabric treated with the nanocomposite with low and high TiO_2_ concentrations, titanium was detected at 5 keV in addition to carbon, oxygen, and sulfur (Table 5). The mass and atomic fractions of oxygen were increased in the cotton fabrics treated with the nanocomposite compared to the cotton fabrics treated with PEDOT:PSS, mainly due to the TiO_2_ structure. In addition, the mass and atomic fractions of titanium in the nanocomposite with 2.22 wt% TiO_2_ were larger than the mass and atomic fractions of titanium in the nanocomposite with 13.29 wt%, with a smaller amount of oxygen; this could explain why the sheet resistance of the conductive cotton with 2.22 wt% was lower than the sheet resistance of the conductive cotton with 13.29 wt%.

### 3.6. XRD Analysis

X-ray diffraction (XRD) analysis was performed to investigate the crystalline structure of the untreated cotton fabric, the cotton fabric treated with PEDOT:PSS, and the cotton fabric treated with the nanocomposite (see Figure 5A). For the untreated cotton fabric, sharp diffraction peaks corresponding to a well-ordered cellulose structure are clearly visible. The diffraction pattern shows peaks at 2θ = 14.7°, 16.5°, 22.7°, and 34.5°, corresponding to the crystal lattices (11$\bar{0}$), (110), (200), and (004) of cotton, which is consistent with the values reported in the literature [26]. The XRD spectra of the cotton fabrics treated with PEDOT:PSS and PEDOT:PSS/TiO_2_ nanocomposite had similar peaks to those of the untreated cotton, which may be explained by the merging of the peaks of PEDOT:PSS or the nanocomposite with the broader peak of the untreated cotton at (200). The XRD pattern of the cotton fabrics treated with the nanocomposite confirmed the tetragonal structure of anatase TiO_2_. It was observed that the intensity of the cotton diffraction peak was reduced in the nanocomposite cotton fabric treated with 13.29 wt% TiO_2_ compared to the untreated sample. This could be due to the agglomeration of TiO_2_ particles when the concentration of TiO_2_ particles increased, resulting in lower conductivity of the samples, which is consistent with the SEM and EDS results.

### 3.7. FTIR Analysis

Figure 5B shows the FTIR analysis of untreated cotton fabric and cotton fabric treated with PEDOT:PSS and the nanocomposite in the wavenumber range 500–4000 cm^−1^. The FTIR spectra of the untreated cotton fabric show distinct peaks at 3308 and 1645 cm^−1^, associated with the pure cellulose structure and attributed to the -OH vibration. The peaks at 2900 and 1041 cm^−1^ can be attributed to the -CH_2_ vibration and the -C-O vibration, respectively. The peaks at 1157 and 1026 cm^−1^ can be attributed to ring breathing and C-O stretching, respectively [49]. The FTIR spectra of cotton fabric treated with PEDOT:PSS show the same peaks as those of the untreated sample, in addition to peaks at 1130 cm^−1^, which can be attributed to S-OH stretching. The nanocomposite-treated cottons show an additional peak at 1024 cm^−1^ that can be attributed to the stretching of the metal oxide bond (O-Ti-O) [50]. The presence of the characteristic peak of TiO_2_ indicated that TiO_2_ was incorporated successfully in the polymer matrix.

### 3.8. Thermal Analysis

The thermal stability of the untreated cotton fabric and the cotton fabrics treated with PEDOT:PSS and nanocomposite was investigated using thermogravimetric analysis (TGA) and differential thermal analysis (DTA). Thermal analysis was performed in a temperature range from room temperature to 900 °C at a constant ramp rate of 50 °C/min, as shown in Figure 6A, and measurements were performed under nitrogen. TGA analysis of the untreated cotton fabric (Curve a) shows a weight loss of 4.97% at the beginning of TGA in the temperature range 30 °C to 100 °C, which is due to the evaporation of moisture in the cotton fabric. Upon further heating to 250 °C, there was no further weight loss, indicating that no adsorbed materials were present. The second weight loss in the TGA curve is due to the degradation of cellulose in the fiber, with a maximum rate of decomposition at 424 °C. It can also be seen that the untreated cotton fabric is thermally stable up to 350 °C. The same trend was observed in the TGA analysis of the cotton fabrics treated with PEDOT:PSS and the nanocomposite (Curves b and c). It can also be seen that the thermal stabilities of the samples are 234 and 278 °C, respectively. Further elevation of temperature above these values results in weight losses of about 55% due to the splitting of the PSS sulfonate groups. From the TGA curves, we conclude that the nanocomposite cotton fabric is more stable than the sample without TiO_2_. The DTA curves shown in Figure 6B indicate that the maximum decomposition rate of the cotton fabrics treated with PEDOT:PSS and the nanocomposite was achieved at 350 °C, with negative peaks indicating an endothermic process.

### 3.9. Washability

The effect of washing on the electrical properties of nanocomposite cotton fabric was studied. The conductive cotton fabric was washed with a steam-washing machine without the use of commercial detergent solutions compared to most conventional methods (see supplemental video, Figure 7A). The steam-washing procedure was repeated for seven cycles, with each cycle lasting 20 min. Natural drying lasted 120 min at room temperature. After each cycle, sheet resistance was measured using the four-probe method. The steam washing study was performed over two days. Cycles 1, 2, and 3 were studied on the first day, and cycles 4, 5, 6, and 7 were studied on the second day. As shown in Figure 7B, the sheet resistance of the sample was stable after three cycles on the first day. However, the measured sheet resistance of the sample increased by 11.48% in cycle 4 on the second day, which could be due to moisture absorption; then, the sheet resistance became very stable in cycles 5, 6, and 7 without losing nanocomposite. Figure 8 shows the image of the cotton fabric before washing, during washing and after washing.

## 4. Conclusions

Flexible conductive cotton fabrics with a smaller amount of conductive materials were developed to obtain highly conductive textiles. The conductive cotton fabrics were prepared by applying PEDOT:PSS/TiO_2_ nanocomposites to the surface of the fabrics by drop-casting, fixing the amount of PEDOT:PSS, and changing the concentration of TiO_2_. The sheet resistance of the conductive cotton fabric can be reduced by 81.41% from 20.71 Ω/□ to 3.85 Ω/□ when the cotton fabric is coated with nanocomposite containing 2.20 wt% TiO_2_. The optimum TiO_2_ concentration was 2.29 wt%, which resulted in a minimum sheet resistance of 2.68 Ω/□. This method allows the production of highly electrically conductive cotton fabrics with a low content of PEDOT:PSS and a low content of TiO_2_ compared to most conventional methods. In addition, the effect of UV light irradiation on the sheet resistance of cotton fabrics coated with nanocomposites was investigated. The results show that sheet resistance depended on the duration of UV light irradiation. The optimum irradiation time was 10 min, during which the sheet resistance of the nanocomposite decreased by 53.41% from 11.87 Ω/□ to 5.53 Ω/□, indicating higher electrical conductivity due to the decomposition of organic chemical bonds in the PEDT:PSS structure, whose binding energy is less than the energy of 365 nm UV light. Modeling of conductivity behavior during heating of the nanocomposite-coated cotton fabrics was performed using the Arrhenius equation for low-, medium-, and high-TiO_2_ concentrations. For the cotton fabrics prepared with nanocomposite with low- and medium-TiO_2_ concentrations, the conductivity increased with increasing temperature, indicating semiconductor behavior; then, a transition to metallic behavior occurred. In contrast, the cotton fabric with the nanocomposite with high-TiO_2_ concentration behaved metallically at all temperatures. The activation energies for the cotton fabrics coated with the nanocomposite were determined by fitting the experimental conductivity curves to this model. The results of steam washing show that cotton fabrics coated with nanocomposites are washable and reusable, which makes them more suitable for electronic wearables. The limitation of our work is that the sheet resistance of the conductive cotton fabrics was studied in a small number of steam washing cycles. Therefore, in our future work, we will closely study the effects of steam washing on electrical properties over a large number of steam washing cycles and use these conductive fabrics to fabricate electrodes for electronic and medical applications.

## Figures and Tables

**Figure 1 materials-16-01738-f001:**
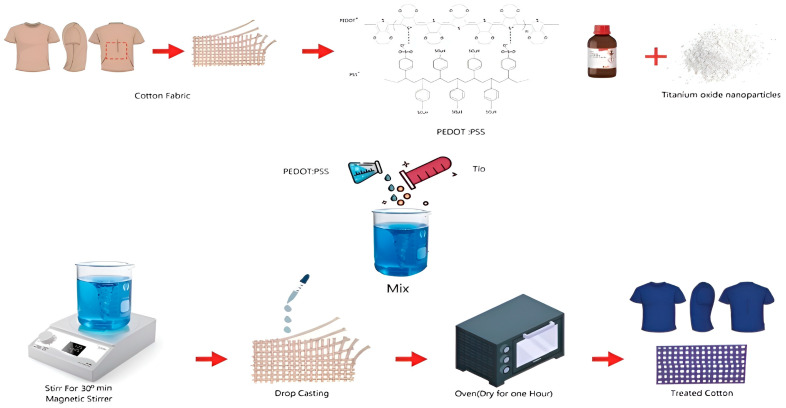
Diagrammatic representation of the fabrication of a conductive cotton fabric based on PEDOT:PSS/TiO_2_ nanocomposite, including the preparation of the mixed solution and the chemical structure of PEDOT:PSS.

**Figure 2 materials-16-01738-f002:**
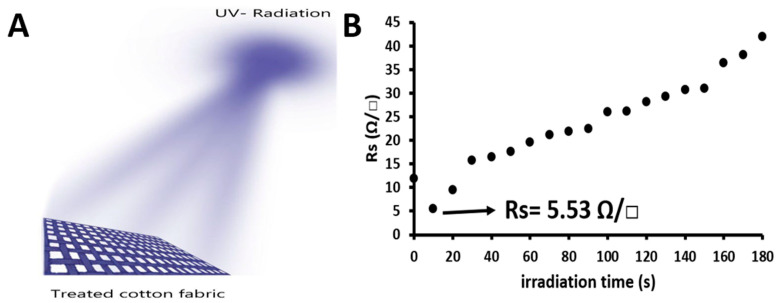
(**A**) Schematic representation of the irradiation of treated cotton fabric with UV light. (**B**) Sheet resistance of nanocomposite cotton fabric as a function of irradiation time.

**Figure 3 materials-16-01738-f003:**
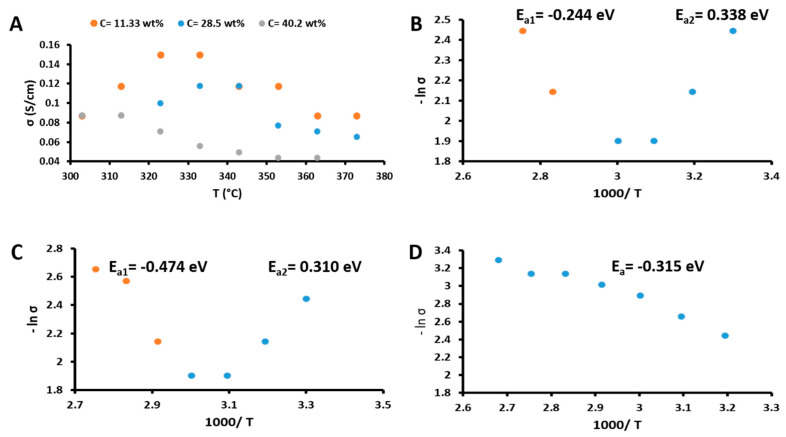
(**A**) Conductivity as a function of temperature. (**B**–**D**) are Arrhenius diagrams for the nanocomposite at low (11.33 wt%), medium (28.5 wt%), and high (40.2 wt%) TiO_2_ concentrations, respectively, and a 9.11 wt% PEDOT:PSS.

**Figure 4 materials-16-01738-f004:**
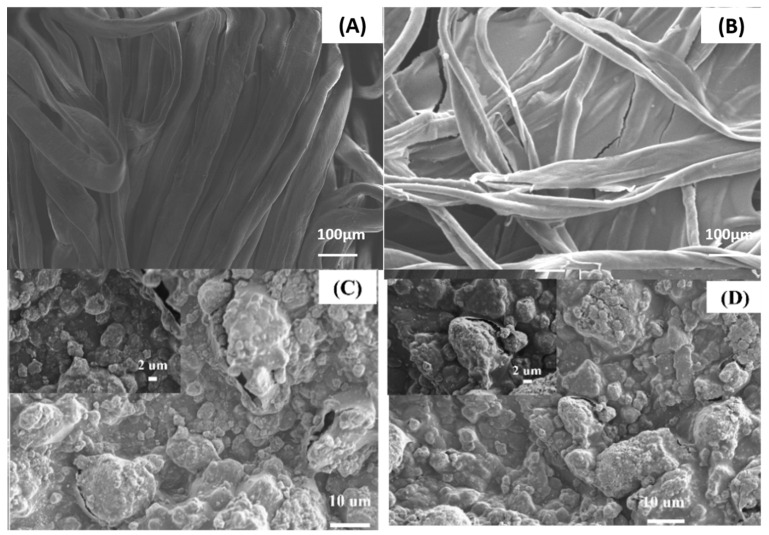
SEM images of (**A**) untreated cotton, (**B**) cotton treated with PEDOT:PSS, (**C**,**D**) cotton treated with nanocomposites containing 2.22 wt% and 13.29 wt% TiO_2_. The insets show highly magnified images of the respective samples.

**Figure 5 materials-16-01738-f005:**
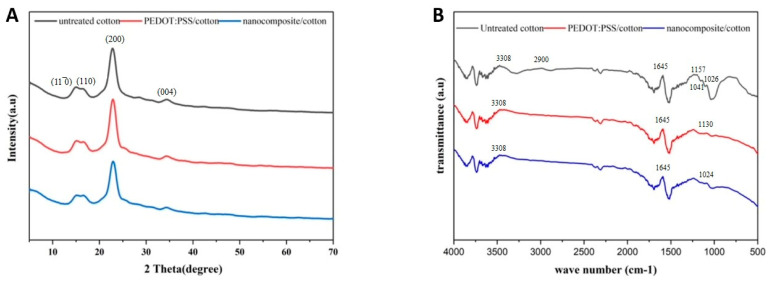
Comparison of XRD analysis (**A**) and FTIR spectra (**B**) of the untreated cotton fabric and cotton fabrics treated with PEDOT:PSS and nanocomposite.

**Figure 6 materials-16-01738-f006:**
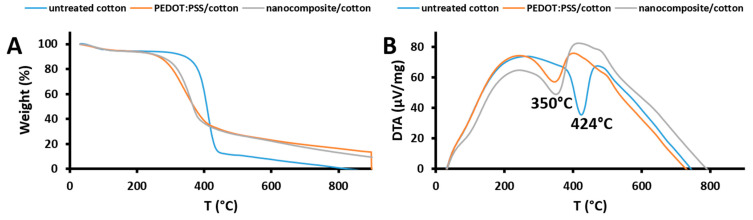
TGA analysis (**A**) and DTA analysis (**B**) of the untreated cotton fabric and the cotton fabrics treated with PEDOT:PSS and the nanocomposite.

**Figure 7 materials-16-01738-f007:**
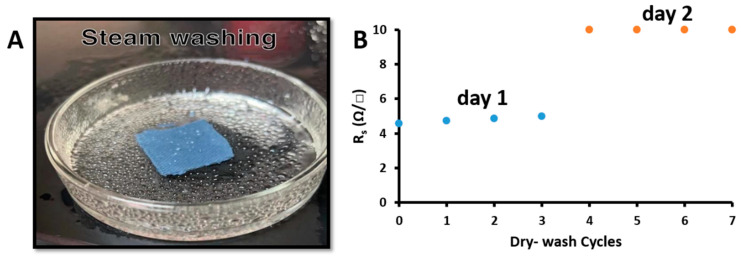
(**A**) Steam washing procedure. (**B**) Conductivity tests of the nanocomposite cotton fabrics after the steam washing procedure.

**Figure 8 materials-16-01738-f008:**
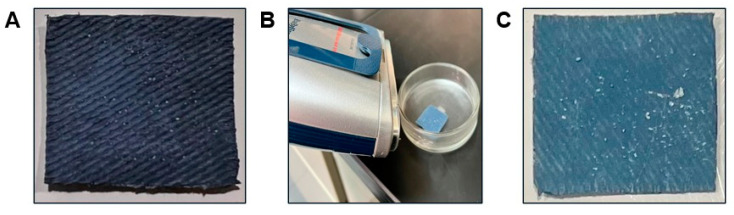
The image of the cotton fabric before washing (**A**), during washing (**B**) and after washing (**C**).

**Table 1 materials-16-01738-t001:** Sheet resistance (Rs) of reference and nanocomposite PEDOT:PSS/TiO_2_ cotton fabrics.

Sample	C(wt%) of the Dope PEDOT:PSS	C(wt%) of TiO_2_	Rs (Ω/□)
1	9.11	0	20.71
2	9.11	2.22	3.85
3	9.11	2.92	2.68
4	9.11	3.39	3.86
5	9.11	5.29	5.46
6	9.11	13.29	13.57

**Table 2 materials-16-01738-t002:** Comparison of sheet resistance values between our work and other reports in the literature.

Substrate	Nanocomposite Materials	Rs (Ω/□)	Ref.
Cotton	PEDOT:PSS without nanoparticles	20.70	This work
films	PEDOT:PSS/MWCNTs	51	[43]
Quartz	PEDOT:PSS/MWCNTs	~500–700	[44]
PET	PEDOT:PSS/SiO_2_	3.20	[45]
Cotton	PEDOT:PSS/TiO_2_	13.57	This work

**Table 3 materials-16-01738-t003:** The binding energy of some chemical bonds in the PEDOT:PSS structure.

Chemical Bond	Bond Energy (kJ/mol)
C–C	347.7
C–S	370
C–O	351.5

**Table 4 materials-16-01738-t004:** Activation energies for the nanocomposite samples.

TiO_2_ Concentration (wt%)	Activation Energy (eV)	
	E_a1_ (303–333 K)	E_a2_ (343–373 K)
2.22	0.3378	−0.2438
19.39	0.3100	−0.4738
31.09	−0.315	

**Table 5 materials-16-01738-t005:** EDS analysis of untreated and PEDOT:PSS-treated cotton fabric.

	Untreated Cotton				Cotton Treated with PEDOT:PSS			
Element Line	E (keV)	M (%)	At (%)	Error	E (keV)	M (%)	At (%)	Error
C-K	0.277	37.73	44.66	±1.37	0.277	50.74	58.79	±1.21
O-K	0.525	62.27	55.34	±2.02	0.525	45.51	39.58	±2.25
S-K	-	-		-	2.307	3.75	1.63	±0.39

**Table 6 materials-16-01738-t006:** EDS analysis of nanocomposite-treated cotton fabric.

	Cotton Treated with Nanocomposite with 2.22 wt% TiO_2_				Cotton Treated with Nanocomposite with 13.29 wt% TiO_2_			
Element Line	E (keV)	M (%)	At (%)	Error	E (keV)	M (%)	At (%)	Error
C-K	0.277	9.34	16.44	±0.95	0.277	15.49	24.78	±1.21
O-K	0.525	48.77	64.46	±2.24	0.525	50.81	61.01	±2.25
S-K	2.307	2.77	1.83	±0.34	2.307	3.52	2.11	±0.39
Ti-K	5	39.12	17.27	±1.03	5	30.18	12.10	±0.91

## Data Availability

All data of this investigation are included in the article.

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
