# Peer review of "The Influence of Titanium Oxide Nanoparticles and UV Radiation on the Electrical Properties of PEDOT:PSS-Coated Cotton Fabrics"

_materials, 2023, doi:10.3390/ma16041738_

Round 1
Reviewer 1 Report
PEDOT:PSS is one of the most promising conductive materials for the production of conductive fabrics due to its excellent properties such as solubility, high conductivity and market availability. I think the author's research has a certain effect on the electronic textile industry, and the author needs to make some improvements before the article is accepted.
1. The introduction should focus on PEDOT:PSS, and the author needs to add more relevant literature.
Section 2.2.1 Materials requires more information, and the nature of materials requires more explanation.
Section 3.3.2 "When the nanocomposite sample was irradiated with UV light for 10 minutes, the sheet resistance of the nanocomposite sample decreased by 53.41% from 11.87Ω /â–¡ to 5.53Ω /â–¡. The sheet resistance of the nanocomposite sample decreased by 53.41% from 11.87ω /â–¡ to 5.53ω /â–¡. . The author needs to explain why.
Section 4.3.4 SEM analysis is not clear enough, and the author needs to add more in-depth analysis. It is suggested that the author make an in-depth analysis from the structure. The author should add details about the relationship between structure and performance.
5. The author has conducted a large number of tests, but generally speaking, the author needs to unify the results of various analyses and add more original analysis.
Author Response
plaese check the attach file

Reviewer 2 Report
The author investigated the influence of titanium oxide nanoparticles and UV radiation on the electrical properties of PEDOT:PSS-coated cotton fabrics. The research topic is interesting, however the presented result are not good enough to be accepted for publication in its current form. I have comments as below:
1. Microstructure and properties of starting materials such as TiO2 should be presented.
2. The experimental section should be rewritten to address what authors want to do. The authors mentioned the TiO2 concentration from 11 to 51 wt.% in the experimental but the TiO2 concentrations in the samples for measuring the resistivity are different.
3. The typical peaks on XRD patterns and FTIR of materials should be indexed.
4. The conductive of the PEDOT:PSS should be measured and presented. The sheet resistance of the fabric coated by PEDOT:PSS is about 20.7 Ohm/sq, this value seem too low compared to the best conductive of the commercial PEDOT:PSS.
5. The stability of the electrical conductivity of samples versus times should be investigated and presented.
6. The presented results should be compared with other reports.
Author Response
plaese check the attach file

Reviewer 3 Report
In this work, the authors studied the effects on the electrical properties due to UV irradiation and nanocomposites based on TiO2-PEDOT:PSS. However, some parts need to be improved before acceptance for publication.
1. The unit of sheet resistance must be corrected throughout the manuscript.
2. Gas sensing and EMI is a little different from this work. This must be removed. Rather focus on different irradiation on PEDOT:PSS to make the readers focus on the importance of this work. Some of the works published recently can be added here in this part.
a. DOI: 10.1039/d1ra03463d
b. https://doi.org/10.1039/C2JM34556K.
3. What is the purity and conductivity of PEDOT:PSS?
4. Change all the units in SI format like minutes to min, etc.
5. On Page 5, the authors mentioned that "the washing procedure was repeated for five cycles". However, in the abstract, it was mentioned to be 7 cycles. Which statement is correct?
6. No DMSO was included n this batch of samples shown in Table 1?
7. On Page 5, below the section of 3.2, PEDOT:PSS was misspelled. Rectify it.
8. Figure numbers are written wrongly in many places. It should be thoroughly checked before resubmitting this manuscript.
9.What does this different C value means in Figure 3a?
10. Is figure 3 d representing the Arrhenius plot for 40.2 Wt?
11. The authors stated, "When the TiO2 concentration increased to a high level, the TiO2 particles agglomerated". How many images are taken? Because on a large scale both the images look not much change, 2.2% seems to be bigger than 13.2 % particle size at 10 um scale. So this statement by the authors looks not consistent with the data.
12. From XRD analysis, only peaks due to cotton were observed. Even after coating cannot see any change in spectra or broaden due to TiO2 or PEDOT:PSS coating. Grazing incidence or wide-angle XRD must be done.
13. The authors stated "It was observed that the intensity of the cotton diffraction peak was increased significantly in the nanocomposite cotton fabric treated with 13.29 wt% TiO2, while the intensity of the TiO2 peaks was decreased". More concrete evidence is needed for proving this statement.
14. The colors used for each condition must be the same throughout the manuscript. Redraw figure 5.
15. Why is the maximum decomposition reduced after the nanocomposite is deposited on the cotton fabric?
16. The image of the cotton fabric before washing during washing and after washing must be reported.
Author Response
plaese check the attach file

Round 2
Reviewer 1 Report
I agr.ee to accept.
Reviewer 2 Report
The manuscript could be accepted in its current form.
Reviewer 3 Report
Accept